# The Effect of Portion Size Interventions on Energy Intake and Risk of Obesity in School-Aged Children: A Systematic Review and Meta-Analysis

**DOI:** 10.3390/nu17182911

**Published:** 2025-09-09

**Authors:** Salma Luthfiyah Sani, Sara Alfaraidi, Yongqi Mu, Gideon Hot Partogi Sinaga, Atul Singhal

**Affiliations:** 1Childhood Nutrition Research Centre, UCL Great Ormond Street Institute of Child Health, London WC1N 1EH, UK; 2Division of Medicine, University College London (UCL), London WC1E 6JF, UK; 3Qassim Health Cluster, Ministry of Health, Buraydah 52367, Saudi Arabia; 4Faculty of Medicine, University of Indonesia, Jakarta 10430, Indonesia

**Keywords:** portion size, energy intake, obesity, school-aged children

## Abstract

**Objectives**: An increase in food portion size offered to children over recent decades has been suggested to contribute to the rise in childhood obesity. This review investigated the effect of interventions that manipulated portion size on energy intake and risk of obesity in school-aged children. **Methods**: A systematic search was performed using MEDLINE, Embase, and Cochrane Library databases (from inception to 2025). Included studies were original articles in English, involving children aged 5–17 years, that focused on portion size interventions using an experimental or controlled study design, with energy intake, body weight, or body mass index (BMI) as the study outcome. The risk of bias was evaluated using the Quality Criteria Checklist (QCC). **Results**: From 514 articles identified, 10 met the inclusion criteria, including a total of 1765 participants. Larger portion sizes increased food intake (grams) and/or energy intake (kcal) in eight studies but did not affect energy intake in one study. Another study focusing on fruit and vegetable portions found inconsistent results. The meta-analysis found that larger portion sizes were associated with higher energy intake compared to the reference portion (mean difference = 86.0 kcal/meal, 95% CI [62.2, 109.9], *p* < 0.00001). **Conclusions**: Offering children larger portions increases energy intake. However, this finding was limited by being based mainly on studies which manipulated portion size at a single meal, in a laboratory setting, and with only short-term measures of energy intake. Future studies need to investigate the long-term effects of portion size interventions on energy intake and risk of childhood obesity.

## 1. Introduction

The rising prevalence of obesity in children is a major public health issue [1]. This increase is particularly high in school-aged children (aged 5–19 years), with the prevalence of obesity increasing from 0.9% to 7.8% in boys, and from 0.7% to 5.6% in girls between 1975 and 2016 [2]. Obesity in children increases the risk of both short- and long-term health complications, including those affecting cardiovascular, respiratory, neurological, musculoskeletal, endocrine, renal, and gastrointestinal systems [3]. Children living with obesity are also at greater risk of bullying and stigmatisation [4] and are five times more likely to develop obesity as adults compared to children without obesity [5].

Although obesity in children is multifactorial and results from the interaction between genetic, perinatal, dietary, environmental, and psychosocial factors [6], changes in food systems that promote an obesogenic environment are recognised to affect its development. For example, there is an increased availability of generally inexpensive, ultra-processed and energy-dense foods and beverages [2]. Additionally, food portion and packaging size have increased considerably in recent decades [7], a strategy used by the food industry to attract customers and encourage purchase [8], but which could encourage higher food consumption. Children may be particularly susceptible to the food and physical environment as they grow older and gain autonomy, strongly impacting their current and future behaviour that may affect the risk of obesity [2].

When offered a larger portion size, children and adults increase their energy intake, a phenomenon known as the portion size effect [9]. For example, systematic reviews in adults [10] and young children [11] found that larger portion sizes were associated with greater energy intake. However, a similar review has not been conducted in school-aged children who may be particularly vulnerable to the portion size effect [12]. Whether larger portions affect the risk of obesity is uncertain. However, a recent systematic review showed associations between portion size and childhood adiposity [13], although in contrast to the current study, this previous review was a narrative synthesis based mainly on observational evidence [13]. Therefore, interventions that reduce portion size have been recommended to recalibrate consumption norms and to help prevent obesity [14]. For example, the WHO suggests limiting portion size in order to reduce the risk obesity [15], and the UK Scientific Advisory Committee on Nutrition (SACN) has recommended the development of age-appropriate portion sizes for foods and beverages to help prevent overconsumption [16].

School age is an important period in shaping health behaviour in children [17], and portion sizes of energy-dense foods and soft drinks have been associated with higher Body Mass Index (BMI) in school-aged children [12]. However, the effect of portion size interventions on food and energy intake has not been systematically investigated in school-aged children (5–17 years). Therefore, the objective of this systematic review was to critically evaluate and summarise the evidence that portion size interventions affect energy intake and the risk of obesity in school-aged children.

## 2. Materials and Methods

This systematic review followed the PRISMA (Preferred Reporting Items for Systematic Reviews and Meta-Analysis) guideline. The protocol was also registered in PROSPERO with registration number CRD42024582043.

### 2.1. Search Strategy

A systematic search was conducted in MEDLINE (Ovid), Embase (Ovid), and Cochrane Library using synonyms and variations in relevant search terms and medical subject headings (MeSH) terms with Boolean operators (NOT, AND, OR). The following search terms were derived using the PICO structure: (1) Population: children aged 5–17 years. (2) Intervention: those targeting portion size or serving size. (3) Comparative group: either standard portion size or no intervention. (4) Outcome: effect of an intervention on dietary intake (in grams), energy intake (expressed as kcal or kJ), body weight, or BMI. The initial search was performed in June 2024 and updated in June 2025 to identify any new publications available, although no new eligible articles were identified. The full search terms for each database can be found in Appendix A.

### 2.2. Inclusion and Exclusion Criteria

The inclusion and exclusion criteria applied to selected studies are presented in Table 1. Studies included were original articles written in English and published from database inception to 2025. Unlike previous reviews, which included observational and cross-sectional studies [13], this systematic review only included experimental studies (randomised controlled trials, quasi-randomised trials, or crossover designs) in order to support a possible causal link between the portion size intervention and study outcomes. Therefore, non-experimental or not controlled studies, cross-sectional studies, review articles, and qualitative studies were excluded because of the low quality of evidence.

### 2.3. Data Collection and Extraction

Search results were exported into EndNote version 20.6, and duplicates were identified and removed. Title and abstract screening were carried out by two independent researchers (S.L.S., Y.M.) at different time points. Full-text screening was performed by applying the inclusion and exclusion criteria. Any discrepancies were resolved by discussion with an additional reviewer (S.A.). Finally, eligible studies were extracted in the standardised form in a results table, which included the following headings: citation, location, study design, study objective, study participants, eating setting, intervention method, intervention duration, outcome evaluation, main finding, strengths, and limitations. The authors of the included studies were contacted if there was missing data or information.

### 2.4. Risk of Bias Assessment

The quality and potential bias of selected studies were critically assessed by two independent researchers (S.L.S., Y.M.) using the Quality Criteria Checklist (QCC) for primary research from the Academy of Nutrition and Dietetics. The checklist is suitable for assessing randomised and non-randomised controlled trials. It includes questions related to the research question, subject selection, comparable groups, withdrawals, blinding, intervention, outcomes, statistical analysis, conclusion support, and likelihood of bias from sponsors. Each question was assessed as “yes”, “no”, or “unclear” to determine whether the study was minus/negative (high risk of bias), neutral (moderate risk of bias), or plus/positive (low risk of bias). A study was marked as having high risk of bias if six or more of the validity questions were “no”, or moderate risk of bias if the answer for any of the first four validity questions was “no”, but other criteria indicated strengths and were answered as “yes”. If most of the answers for the validity questions were yes (and included ‘yes’ to questions 1–4), the study was regarded as having low risk of bias [18].

### 2.5. Synthesis of Results, Heterogeneity, and Reporting Bias Assessment

Random-effects meta-analysis with inverse-variance weighting was performed with Review Manager (RevMan) 5.4 and R with “meta” package. Results were presented as the mean difference with 95% confidence intervals (CI) in energy intake (kcal) between portion size conditions (reference vs. larger portion offered). Subgroup analyses were also conducted to examine if the portion size effect differed between laboratory and natural eating settings. Heterogeneity was assessed using the *I*^2^ statistical test (>50% value was indicated as substantial heterogeneity) and statistical significance was set at *p* < 0.05. Furthermore, funnel plots and Egger’s test were conducted to address possible publication bias, with *p* < 0.05 to be considered as statistically significant.

## 3. Results

### 3.1. Results of Study Selection

The selection process of articles to meet the eligibility criteria is shown in Figure 1. Search results from three databases (MEDLINE, Embase, Cochrane Library) yielded a total of 514 articles. After removing duplicates, 394 articles were screened based on titles and abstracts. Forty-five relevant articles were selected and further assessed against the study inclusion and exclusion criteria. Three studies were not retrieved because only abstracts, and not full manuscripts, were available. The main reasons for article exclusion were not focusing on a portion size intervention (*n* = 11), observational study design (*n* = 3), being review articles or protocols (*n* = 6), and study participants not being aged 5–17 years (*n* = 12). Ten articles met all study eligibility criteria and were included in this systematic review [19,20,21,22,23,24,25,26,27,28].

### 3.2. Results of Quality Assessment

Based on the QCC for primary research [18], nine studies had a positive quality rating, which indicates a low risk of bias. As a result of unclear inclusion/exclusion criteria for subject selection, one study had a neutral rating suggesting a moderate risk of bias [28] (Figure 2). The Robvis Tool was used to visualise the risk of bias in the included studies [29].

### 3.3. Characteristics of Studies

A summary of the studies included in this review is presented in Table 2. Eight studies were conducted in the USA [19,20,21,22,23,24,25,26], one study in Australia [27], and one study in the UK [28]. Three studies had a randomised crossover design [19,23,25], three studies had a between-subjects design [20,22,28], three studies had a within-subjects design [21,24,26], and one study was a population-based randomised trial [27]. The sample size varied from a small group intervention (*n* = 18) [19] to a large population study (*n* = 1299) [27], but most studies had a sample size between 40 and 100 children. Along with including children aged 5–17 years, one study also involved pre-school children [20], and one study involved parents [27].

All studies only evaluated the portion size effect at a single meal time. Five studies used a portion size intervention at dinner time [20,21,23,25,26], two studies at lunchtime [19,24], two studies at snack time [22,27], and one study for breakfast [28]. Regarding the type of food changed in the portion size intervention, three studies varied the offered portion size of full set meals [24,25,26], two studies varied only the portion size of the main entrée (macaroni and cheese) as part of a full set menu [20,21], two studies varied the portion size of energy-dense snacks [22,27], one study varied the portion size offered of a fast-food meal [19], one study varied the portion size of fruits and vegetables within a full set meal [23], and one study manipulated the front of pack visual on a packet of breakfast cereal [28]. Regarding the intervention setting, two studies were conducted in a natural eating setting of a food court [19] or a school classroom [24], and the rest were conducted in a laboratory setting.

### 3.4. The Effect of Portion Size Intervention on Energy Intake in School-Aged Children

Most studies (8 of 10) in this systematic review found that a portion size intervention had a significant effect on food consumption and energy intake in school-aged children. Six studies found that offering larger portions resulted in a higher energy intake [20,21,24,25,26,27], and two studies found that exposure to larger portion sizes resulted in a higher food intake (expressed in grams) [22,28]. One study found that doubling the portion sizes of fruits and vegetables offered within a set meal decreased the energy density of the meal (by 0.06 ± 0.02 kcal/g) and increased fruit intake, but not vegetable intake [23]. Only one study, based on altering portions and eating rate of a fast-food meal, found that larger portion sizes did not lead to a statistically significant effect on energy intake [19]. Because of a lack of longitudinal data collection or longer-term follow-up, none of the studies evaluated the effect of portion size intervention on body weight or BMI as the study outcome.

Eight out of ten studies investigated the effects of children’s characteristics on their response to portion size interventions, such as the child’s weight, age, and sex. Eight studies found that children’s weight status was not significantly associated with the change in energy intake with portion size variation (i.e., both children living with obesity and children with normal weight consumed higher energy intake when exposed to larger portions) [20,21,22,23,24,25,26,27]. Two studies found that the children’s age was not associated with the change in energy intake with a larger portion size [20,21]. The effect of sex on the portion size effect was inconsistent: two studies showed that sex was not significantly associated with the change in energy intake with portion size change [21,24], while two other studies found that boys consumed significantly higher energy than girls when offered larger portion sizes [23,27]. Other factors found to be associated with changes in food and energy intake with different portion sizes, included food insecurity [24], food preference [23], food responsiveness, and satiety responsiveness [26]. One study evaluated the children’s perception about different portion size images and found that 63% of children perceived the image of portion as appropriate regardless of whether the image was of a large or a small portion size [28].

### 3.5. Results of Meta-Analysis

The meta-analysis included 18 comparisons from nine studies that analysed the mean difference in total energy intake (kcal) between smaller and larger portions (Figure 3). One study was excluded from the meta-analysis because the data on energy intake was not provided [28]. Overall, a larger portion size was associated with increased energy intake by 86.0 (95% CI [62.2, 109.9]) kcal at a meal compared to the reference or smaller portion size (*p* < 0.00001). However, there was high heterogeneity of the studies (*I*^2^ = 91%).

Subgroup analysis was performed to examine if results differed between laboratory and natural eating settings. In a laboratory setting, the pooled effect estimate demonstrated a significant positive portion size effect (mean difference between larger and reference portion size groups = 86.8 kcal, 95% CI [61.8, 111.7], *n* = 7 studies, *p* < 0.00001). This effect was similar to a natural eating setting, (mean difference between portion size groups = 83.7 kcal, 95% CI [28.5, 138.9], *n* = 2 studies, *p* = 0.003). Comparison of laboratory and natural settings did not statistically differ (*p* = 0.9).

### 3.6. Results of Reporting Bias Assessment

The potential for publication bias was assessed by generating funnel plots of nine studies included in the meta-analysis, where symmetry can be observed (Figure 4). The test for asymmetry from Egger’s test was not statistically significant (t = −0.9, *p* = 0.3), which supports a lack of publication bias.

## 4. Discussion

### 4.1. Summary of Main Findings

This systematic review investigated the effect of portion size interventions on energy intake and body weight in children aged 5 to 17 years and identified 10 relevant studies. Eight of these ten intervention studies found that larger portion sizes led to increased food intake (in grams) and greater energy intake by a mean of 86 kcal/meal compared to the reference portion. This result is consistent with a previous systematic review, which showed that larger portion sizes were associated with increased energy intake in young children aged 2 to 12 years (186 kcal/day) [9] and in adults (295 kcal/day) [10]. A larger portion size was not significantly associated with high energy intake in one included study, possibly because of the small study sample size [19]. Another study that focused on a portion size intervention for fruit and vegetable intake showed that doubling fruit and vegetable intake led to lower energy density of the meal but did not affect the total energy intake [23]. Due to the lack of longer-term follow-up, insufficient evidence was available to determine if portion size interventions could affect children’s overall or habitual energy intake or lead to changes in body weight and the risk of obesity.

### 4.2. The Portion Size Effect on Energy Intake in School-Aged Children

Although further research is required to investigate if children compensate for over-eating at one meal by eating less in subsequent meals, the 86.0 kcal increase in energy intake per meal, when children are offered a larger portion compared to the reference, if sustained, may contribute to excess weight gain. For example, a previous longitudinal study in children aged 5–7 years found that there was only a 69–77 kcal/day difference in energy intake between children who put on weight versus those who did not [30]. Similarly, a longitudinal study in children aged 6–14 years suggested that to prevent overweight, excess energy intake should not exceed 46–53 kcal/day in boys and 58–72 kcal/day in girls [31]. Furthermore, a mathematical model developed from the US population suggests that a mean increase in energy intake of 190 kcal/day in girls, and 210 kcal/day in boys, contributes to excess weight gain [32].

The effect of larger portion size on energy intake suggested in our review was greater than the effects of other factors affecting the food environment. For example, in an intervention study, children exposed to multiple-media food advertising increased daily energy intake by 46 kcal (194 kJ, 95% CI [80, 308], *p* = 0.001) compared to those exposed to non-food advertisements [33]. Thus, our finding of 86.0 kcal/meal increase in energy intake with larger portion sizes could potentially lead to increases in weight gain, assuming that a single meal contributes to 30% of daily energy intake and children do not compensate with lower energy intake following consumption of the larger portion. Importantly, the effect of larger portion sizes was similar in natural and laboratory settings (the latter minimises confounding factors that affect eating behaviour). This observation suggests that portion size interventions may be effective in real-world environments, although with only two studies in natural settings, further work is needed in this area.

Several mechanisms could explain the effect of larger portion sizes on energy intake. First, the most profound mechanism is unit bias, which suggests that people may see one serving as an appropriate amount to consume irrespective of its size [34]. For example, in this review, children consumed higher amounts of food when a larger portion size of dinner meal was served [20,21,25,26] or when more variety of snacks was offered [27]. Second, using different dishware sizes might create visual bias, which is known as the Delboeuf illusion. The Delboeuf illusion affects our perception of food portions, making the same amount appear larger on smaller plates and smaller on larger plates, potentially influencing how much we eat [35]. This phenomenon can be seen from a study of elementary school children who consumed higher energy intake when using adult-size compared to child-size dishware [24]. Third, portion size could set consumption norms that become the reference for how much should be consumed [35]. For example, the study which varied portion size images on cereal boxes showed that children consumed more cereal when exposed to large portion size visual cues compared to the smaller portion size image. Lastly, peer influence could be an important factor that influences portion size in school-aged children. For example, one of the included studies demonstrated that, when preadolescent girls were exposed to a video in which a model consumed a large serving size of cookies, they themselves consumed more cookies than participants exposed to a video in which the model consumed smaller-sized servings [22]. Therefore, social modelling could shape people’s food choices and intake through the norms provided by others [36].

Importantly, the type of foods offered may lead to different effects of a portion size intervention. Most studies in this review varied portion sizes of foods such as main entrée (macaroni and cheese, pasta, chicken nuggets), breakfast cereal, cookies, and snacks, which are relatively high in energy density and palatability. One study that varied fruit and vegetable portion size showed no effect in increasing vegetable intake. Thus, assessment of the portion size effect should consider aspects such as food energy density and palatability, which could reinforce eating behaviour. The energy density of food can affect how much volume a person perceives they are consuming, which influences satiety and energy intake [37]. People tend to eat with a consistent volume of food [37] so that when the energy density is higher, the same portion size will provide more calories despite similar feelings of satiety. Thus, energy density is positively associated with energy intake [38]. Moreover, high-calorie foods typically have higher palatability than low-calorie foods [37]. Palatable foods can activate hedonic motivational pathways, with a stronger influence on intake when satiated than when hungry, suggesting that hedonic mechanisms could promote intake although homeostatic needs are met [39].

In accordance with the type of food influencing the portion size effect, varying portion sizes of fruits and vegetables offered to school-aged children showed inconsistent effects on fruit and vegetable consumption. Doubling the portion size of fruits and vegetables in a dinner meal increased the intake of fruit (applesauce) but not vegetables (broccoli and carrot) [23]. This finding is similar to an experimental study in a school cafeteria, where a 50% increase in fruit and vegetable portion size in a lunch meal resulted in a higher percentage of students consuming fruits (orange) but not vegetables (carrot) [40]. However, a study across 5 days in preschool children showed that increasing fruit and vegetable portion by 50% with 12 types of vegetables (added with a small amount of butter) and 9 types of fruits could increase both fruit and vegetable intake and decrease daily energy intake [41]. Similarly, in adults, increasing vegetable (broccoli with butter seasoning) portion size by 50% and 100% led to increased vegetable intake [42]. Thus, optimising fruit and vegetable intake and preference in school-aged children may require strategies beyond increasing portion size, such as enhancing palatability, offering variation, and providing repeated exposure, especially when fruits and vegetables are served alongside highly palatable foods.

### 4.3. Factors Affecting Susceptibility to Larger Portion Sizes

Children are vulnerable to the portion size effect irrespective of weight status, as consistently shown in eight studies in the current review [20,21,22,23,24,25,26,27]. This finding is consistent with a study in adults, which showed that response to portion size at a single meal time was not affected by the participants’ BMI [43]. However, a study in preschool children over a 5-day period found that children with a higher BMI-for-age percentile had greater increases in their intake when served larger portions compared to children with a lower BMI-for-age percentile, even after adjustment for energy requirements [44]. This finding could reflect the possibility that children with a higher BMI were not eating more when offered in larger portions simply because of their higher energy requirements, but there might be an interplay between portion size and appetitive traits [44]. In school-aged children, lower satiety responsiveness and higher food responsiveness were associated with higher total energy intake, although the portion size effect did not vary with weight status [26]. Thus, while the portion size effect was generally consistent across weight status in children, there might be differences in individual appetitive traits that influence the response to different portion sizes, a hypothesis requiring further investigation. Furthermore, whether weight status affects the variability in any compensatory response to larger portions also requires further research.

The portion size effect was not associated with age in the current review of school-aged children (in two studies) [20,21]. Previously, it was suggested that preschool children were able to self-regulate energy intake when presented with larger portion sizes in laboratory conditions, but more recently, it has been shown that serving larger portion sizes promoted higher intake in children as young as 2 years old [20]. Conversely, a Chinese study found that there was a difference in response to portion sizes between children aged 4 and 6 years, where older children showed higher increases in intake with increased portion sizes (23.7% more food intake compared to the normal portion), while younger children consumed 29.5% less food with larger portion size compared to the normal portion [45]. The different response in younger children was possibly caused by younger children becoming “overwhelmed” when presented with a larger portion, and thus they consumed less food, although more research is needed to validate this finding [45]. Similarly, a study in Singapore showed that older children (6 years) had a greater response to larger portion sizes of lunch food compared to younger children (3 years), who tended to consume a similar amount of food across serving sizes [46]. Although further research is required, most studies suggest that children increasingly respond to portion size cues with age, and social context becomes more important in influencing eating behaviour as they get older [45].

Children’s sex might influence the susceptibility to the portion size effect. Previously, it has been suggested that because boys may have larger portions due to higher energy requirements than girls, and girls tend to be more aware of their diet particularly as they get older, there may be interactions between children’s sex and the portion size effect [47]. In our review, two studies found that boys consumed significantly higher energy intake than girls when offered larger portion sizes [23,27]. Similarly, a cross-sectional study (and hence not included in this review) showed that sex was a significant predictor of portion size effect, where boys consumed more snacks than girls (especially for chocolate and confectionery) when offered larger portion sizes [47]. However, two other studies in this review found that children’s sex was not significantly associated with the portion size effect [21,24]. Thus, the strategy of downsizing snacks could be targeted at boys in particular [47], but both sexes seem vulnerable to increased intake when served with larger portions.

Socioeconomic status might be associated with the portion size effect in children. In the current review, one study showed that food insecurity was a predictor of response to adult-sized dishware, where children from food-insecure families showed higher increases in total self-served energy (kcal) than children from food-secure families [24]. This could be attributed to the encouragement of eating from parents when food is available, such as during periods when they receive food assistance benefits [24]. Similarly, a cross-sectional study demonstrated that children from lower-income families consumed higher total energy intake of snacks than children from higher-income families, which may be explained by the greater accessibility, affordability, and convenience of snacks for lower-income families [47]. However, another study included in the current review showed that children’s socioeconomic status was not associated with food and energy intake across varied portion sizes [27]. Similarly, a study in adults showed that the effect of portion size on energy intake was not different in participants from lower and higher socioeconomic backgrounds [48]. Thus, although portion size interventions could be targeted to children from lower social economic status, given the higher prevalence of obesity in this population [49], children are likely to be susceptible to portion size effect irrespective of their socioeconomic status.

## 5. Strengths and Limitations

### 5.1. Strengths and Limitations of Included Studies

Studies in this review have several strengths, such as a randomised, experimental, or controlled study design that strengthens the likelihood of a causal link between the portion size intervention and study outcome. Furthermore, all studies conditioned study participants before the intervention, such as abstaining from foods and drinks 2 h before testing and familiarising with the study process. Reference portions and types of food for intervention were also adjusted based on children’s age and habitual intake. However, all studies in this review only evaluated the effect of portion size interventions at a single meal and therefore the long-term effect on energy intake cannot be determined. Nevertheless, previous studies in adults and young children have shown that the portion size effect was sustained over several days, where increased energy intake was not compensated by a lower intake over the following days [44,50,51]. Although children may regulate energy intake over time [7], a portion size intervention over 5 days in children aged 3–5 years resulted in 18% greater daily energy intake [44]. The inability to adjust energy intake after prolonged exposure to larger portion sizes could lead to a sustained overconsumption and an increased risk of obesity [44], a hypothesis requiring further investigation in school-aged children.

Other limitations of studies included in our review are, first, that to minimise confounding factors affecting eating behaviours, eight out of ten studies were conducted in laboratory settings, while only two studies were conducted in natural settings. However, the portion size effect was similar in both settings. Second, there was high heterogeneity between studies because of many different portion size interventions used, the different foods manipulated, and different participant populations. Third, the relatively small sample size and limited variation in age, ethnicity, and baseline weight status limits the applicability of findings to the general population. Fourth, most studies were conducted in the USA, with only one in the UK and one in Australia. Thus, differences in culture and food environment (e.g., frequency and timing of eating, and availability and consumption of highly palatable snack foods) could limit the generalisability of findings to other regions. This is especially important because few portion size interventions have been conducted in low- or middle-income countries which face a rapid increase in obesity rates. Finally, no study has investigated portion size interventions’ effect on body weight and, hence, the risk of obesity.

### 5.2. Strengths and Limitations of This Review

To the best of our knowledge, this review is the first systematic review to synthesise evidence for the effect of portion size interventions on energy intake in school-aged children, and it adds to previous evidence for the portion size effect on energy intake in preschool children and adults [10,11]. This review also had a robust and comprehensive search strategy using three databases, in which all studies were screened and critically appraised by two independent researchers to minimise selection bias. We also found non-significant publication bias through the funnel plot and Egger’s test result. However, there are some important limitations. The search strategy for this review was limited to peer-reviewed publications available in English and did not look for grey literature, which means publication bias could not be excluded entirely. Furthermore, the included studies had different study designs, intervention methods, and outcome measures leading to high heterogeneity between studies in the meta-analysis.

## 6. Conclusions

Our systematic review suggests that larger portion sizes lead to higher energy intake in school-aged children with an effect size (86 kcal/meal) that may contribute to increase in energy intake and, hence, the risk of obesity. Children appear susceptible to the portion size effect regardless of weight status, although further investigation is required to understand the influence of age, sex, and socioeconomic status. Furthermore, future research needs to incorporate repeated measures and longer follow-up periods to assess sustained effects on habitual energy intake and long-term changes in BMI. While the generalisability of the evidence should be interpreted cautiously, given the limitations of previous research in this area, this review suggests that portion size interventions could help reduce the risk of obesity in school-aged children.

## Figures and Tables

**Figure 1 nutrients-17-02911-f001:**
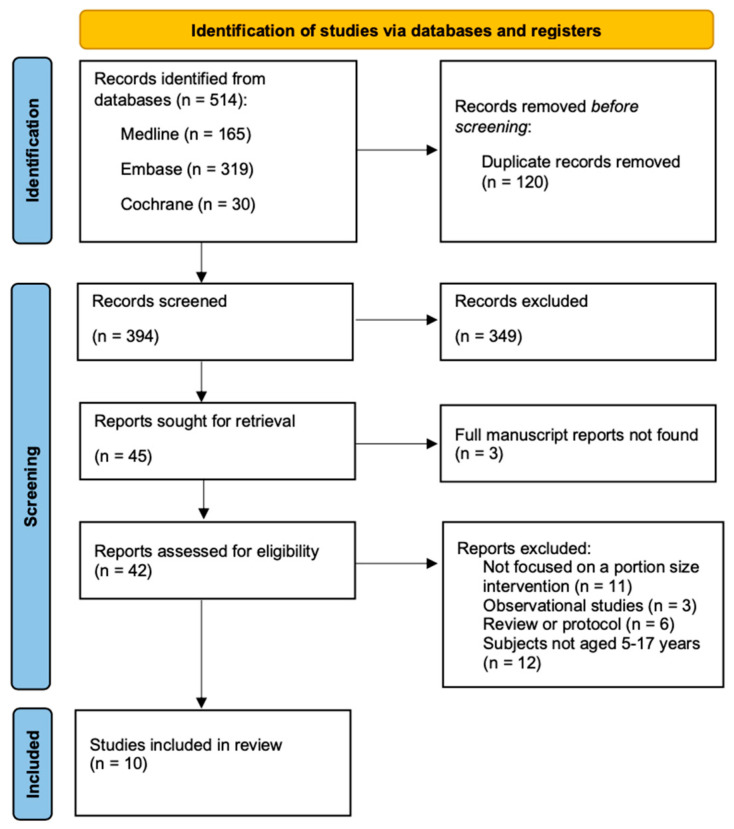
PRISMA flowchart for study selection.

**Figure 2 nutrients-17-02911-f002:**
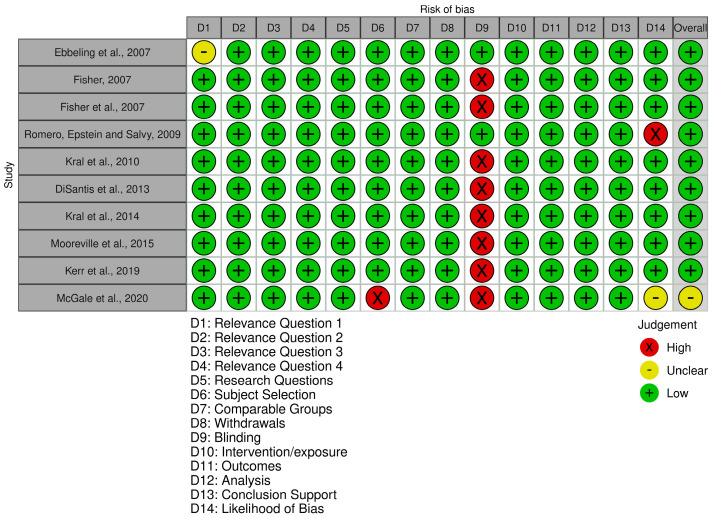
Results of quality assessment of the included studies [19,20,21,22,23,24,25,26,27,28].

**Figure 3 nutrients-17-02911-f003:**
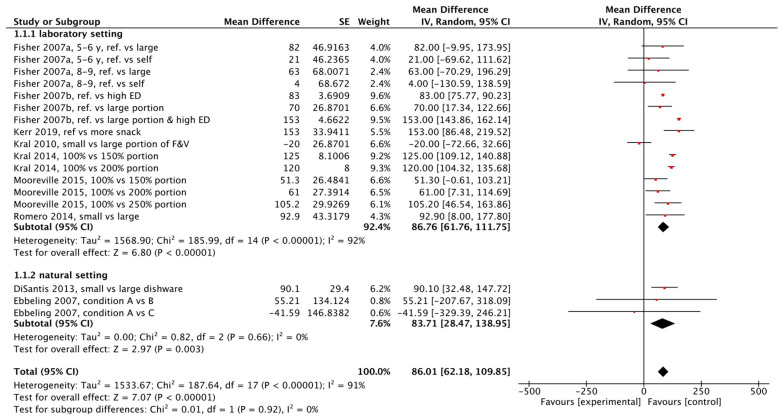
Meta-analysis of mean difference (kcal) in total energy intake between small (reference) and larger portion size [19,20,21,22,23,24,25,26,27].

**Figure 4 nutrients-17-02911-f004:**
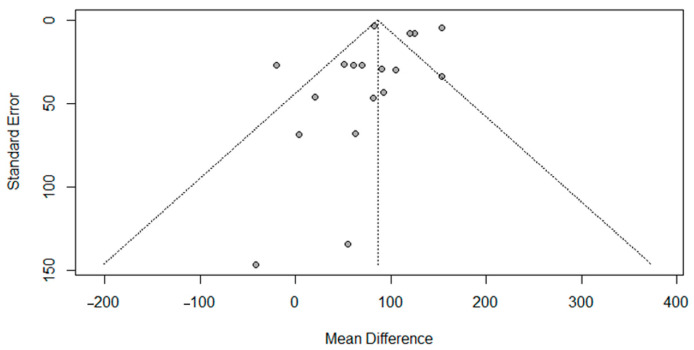
Funnel plot for nine studies reporting mean difference (kcal) between reference and larger portion size. (Test for asymmetry, *p* = 0.3).

**Table 1 nutrients-17-02911-t001:** Inclusion and exclusion criteria of the studies.

	Inclusion Criteria	Exclusion Criteria
Population	School-aged children (5–17 years old)	Pre-school children (<5 years) or adults
	Children without underlying medical conditions	Children with other diseases
Intervention	Focus on the interventions that affect portion size	No intervention on portion size variation
Outcome	The outcome variable was the quantitative assessment of dietary intake (gram), energy intake (kcal or kJ), body weight (kg), or body mass index (BMI)	The outcome variable was not the quantitative assessment of dietary intake, energy intake, body weight, or BMI
Study Types	Experimental trials (randomised), controlled studies, quasi-randomised trials, or a crossover design	Non-experimental or non-controlled studies, reviews, or qualitative studies
	Original articles written in English	Original articles written in other languages

**Table 2 nutrients-17-02911-t002:** Result summary of portion size effect on energy intake in school-aged children.

Citation, Title, Location, Design	Objective, Sample Size, Intervention Method, Eating Setting	Intervention Period, Outcome Evaluation, Follow-Up	Main Findings ^1^	Strengths and Limitations
① Ebbeling et al., 2007 [19] **Title:** “Altering portion sizes and eating rate to attenuate gorging during a fast-food meal: Effects on energy intake” **Location:** Boston, Massachusetts **Design:** Randomised crossover design	**Objective:** To evaluate whether reducing portion sizes and slower eating rate can decrease energy intake when children are offered a fast-food meal. **Sample size:** 18 adolescents (14 female, 4 male) aged 13–17 years who reported consuming fast food ≥1 x per week. **Intervention method:** Fast food meals (chicken nuggets, French fries, ketchup, sweet and sour sauce, and cola) were presented in 3 conditions at different time points: 1 large serving (condition A), portioned into 4 smaller servings (condition B), and portioned into 4 smaller servings served at every 15 min intervals (condition C). The total amounts of foods and beverage given was the same over the study period (5691 kJ). **Eating setting:** Lunch meal in food court.	**Intervention period:** 4 sessions over the summer of 2005 (exact period not given). **Outcome evaluation:** Energy intake (kJ) and meal size rating of each meal. **Follow-up:** Dietary and physical activity data were collected by telephone 2 days after each visit.	Children consumed different amounts of energy in the 3 different conditions (mean ± SEM: condition A = 5552 ± 357 kJ; condition B = 5321 ± 433 kJ; condition C = 5762 ± 500 kJ), but the difference was not statistically significant (*p* = 0.5). However, they consumed ~5460 kJ (~50% of energy needs) under all conditions which indicates that the children were overeating.	**Strengths:** (1) Natural eating setting which minimised confounding factors; (2) Participant conditioning before intervention: eating a standard breakfast of cereal and cold milk, abstaining from foods and drinks until the time of the study visit; (3) Blinding of the interviewer. **Limitations:** (1) Small sample size which limits generalisability of findings; (2) Only evaluated 1 combination of fast-food items; (3) Homogenous study sample (majority were of black ethnicity).
② Fisher, 2007 [20] **Title:** “Effects of age on children’s intake of large and self-selected food portions” **Location:** Houston, Texas **Design:** Between-subjects design (age group) with a within-subject component (portion size). Each child in three age groups was randomly assigned into three portion size sequences.	**Objective:** To determine the effects of age on children’s responsiveness to large and self-selected portions. **Sample size:** 75 children (44 boys, 31 girls) in 3 age groups: 2–3, 5–6, and 8–9 years old. **Intervention method:** Children were served with main entrée of macaroni and cheese that varied across 3 portion sizes: a reference condition, a large condition (portion was doubled), and a self-selected condition (portion was doubled but served in an individual serving dish). Portion sizes of other foods and drinks were held constant. **Eating setting:** Dinner meal in “USDA Children’s Nutrition Research Center’s Children’s Eating Laboratory”.	**Intervention period:** Once a week for 3 weeks **Outcome evaluation:** Energy intake (kcal) of each meal, children’s comment on portion size, children’s bite size and frequency. **Follow-up:** No follow-up (intake was estimated immediately after each meal).	(1) On average, children consumed 13% more total energy (*p* < 0.01) in the larger portion size compared to reference condition. This finding was consistent across the age groups (2–3 y = 294 ± 123 vs. 276 ± 135 kcal; 5–6 y = 562 ± 179 vs. 480 ± 128 kcal; 8–9 y = 700 ± 282 vs. 637 ± 190 kcal). (2) In the self-selected condition, children consumed 11% less of total energy intake (*p* < 0.01) compared to the large-portion condition. However, total energy intake in self-selected condition remained higher than the reference (2–3 y = 280 ± 134 vs. 276 ± 135 kcal; 5–6 y = 501 ± 179 vs. 480 ± 128 kcal; 8–9 y = 641 ± 286 vs. 637 ± 190 kcal). (3) Age, weight status, and sex, did not influence their energy intake in the large-portion condition.	**Strengths:** (1) Relatively large sample size; (2) Clear age grouping; (3) Portions are age-appropriate; (4) Participant conditioning before intervention: abstaining from foods and drinks for prior 2 h. **Limitations:** (1) Conducted in a laboratory setting; (2) No follow-up data collected; (3) 14 children were excluded from the analysis because of zero intake of the main entrée; (4) Homogenous study sample (all children were non-Hispanic white).
③ Fisher et al., 2007 [21] **Title:** “Effects of portion size and energy density on young children’s intake at a meal” **Location:** Houston, Texas **Design:** 2 (portion size) x 2 (energy density) within-subjects factorial design. Randomisation of feeding condition sequences.	**Objective:** To investigate the effects of entrée portion size and energy density on satiation in children. **Sample size:** 53 children (28 girls, 25 boys) aged 5–6 years old. **Intervention method:** Children were served a single meal with macaroni and cheese as main entrée in 4 conditions with variation in the portion size (250 or 500 g), and energy density (1.3 or 1.8 kcal/g). Other foods were constant. **Eating setting:** Dinner time in “USDA Children’s Nutrition Research Center’s Children’s Eating Laboratory”.	**Intervention period:** Once a week for 4 weeks. **Outcome evaluation:** Food intake (g) and energy intake (kcal) of each meal. **Follow-up:** No follow up (intake was estimated immediately after each meal).	(1) Children consumed 15% higher energy when the larger portion size was served compared to the reference portion size (548 ± 19 vs. 478 ± 19 kcal, *p* < 0.001). Similarly, children had 18% greater total energy intake when the high energy density meal was served compared to the reference (554 ± 19 vs. 471 ± 19 kcal, *p* < 0.0001). A combination of larger portion size and high energy density resulted in 34% greater total energy intake than the reference (598 ± 24 vs. 445 ± 24 kcal). (2) Sex, age, and weight status were not significantly associated with portion size and energy density.	**Strengths**: (1) Familiarisation of laboratory setting before testing; (2) Children’s conditioning before intervention: abstaining from foods and drinks for the prior 2 h; (3) Food quantitative sensory tests before testing; (4) Diverse ethnicity of study participants. **Limitations:** (1) Small age range of children; (2) Conducted in laboratory setting; (3) No follow-up data collected.
④ Romero, Epstein, and Salvy, 2009 [22] **Title:** “Peer Modelling influences girls’ snack intake” **Location:** New York, NY, USA **Design:** 2 (weight status) × 2 (portion size) between-group factorial design. Randomisation of participants to intervention condition.	**Objective:** To assess how peer modelling influences eating in preadolescent girls with normal weight and overweight. **Sample size:** 44 girls (22 overweight, 22 normal weight) aged 8–12 years. **Intervention method:** Portion sizes were varied by showing a video in which a model consumed either a small (29 g/1 oz, 10 bite-sized cookies) or a large (223 g/8 oz, 77 bite-sized cookies) serving. Each participant was provided with the same amount of cookies (8 oz). 1 cookie (2.9 g) = 14.03 kcal. **Eating setting:** Snack meal in laboratory.	**Intervention period:** One visit (exact period not given) **Outcome evaluation:** Cookie intake (gram), hunger, and food-liking rating. **Follow-up:** No follow up (intake was estimated immediately after each meal).	Girls with overweight consumed significantly more cookies than girls with normal weight (60.5 ± 35.1 g vs. 41.7 ± 23.2 g; *p* < 0.05). Furthermore, girls exposed to the small portion size condition consumed fewer cookies than girls exposed to the large portion size (41.5 ± 27.2 g vs. 60.7 ± 32.0; *p* < 0.05). However, the interaction of weight status by portion size condition was not statistically significant (*p* = 0.2) which indicates that video model affects both girls with normal weight and those with overweight.	**Strengths:** (1) Comparison of individual variance (weight status); (2) Children’s conditioning before intervention: abstain from food and drinks for 2 h; (3) Blinding of the experimenter. **Limitations:** (1) Homogenous sample (all females, most were white); (2) Eating time limitation (10 min) for participant might affect eating behaviour; (3) Not involving a “no video” comparison group; (4) Not assessing energy intake; (5) Only analysing 1 type of food.
⑤ Kral et al., 2010 [23] **Title:** “Effects of doubling the portion size of fruit and vegetable side dishes on children’s intake at a meal” **Location:** Philadelphia, Pennsylvania **Design:** Randomised crossover design	**Objective:** (1) To evaluate the impacts of doubling the fruit and vegetable portion size on children’s fruit and vegetable consumption; (2) To determine how fruit and vegetable portion size variation affects total meal energy intake. **Sample size:** 43 children (21 girls, 22 boys), aged 5–6 years. **Intervention method:** Children were served dinner meal which consisted of pasta with tomato sauce, fruit and vegetable side dishes (broccoli, carrots, and applesauce), and milk. The fruit and vegetable portion size was doubled between the experimental conditions, while other foods remained constant. **Eating Setting:** Dinner time in “Centre for Weight and Eating Disorders at the University of Pennsylvania”.	**Intervention period:** Once a week for 2 weeks. **Outcome evaluation:** Food intake (g), energy intake (kcal), and energy density (kcal/g) at each meal; taste preference. **Follow-up:** No follow up (intake was estimated immediately after each meal).	(1) Doubling the portion size of fruit and vegetable side dish increased the intake of apple sauce (43%, 36.1 ± 9.9 g, *p* = 0.001), but not broccoli (*p* = 0.7) and carrots (*p* = 0.6). There was no significant interaction between weight status and fruit and vegetable portion size. (2) The difference in total meal energy intake between doubled fruit and vegetable portion and reference was not statistically significant (446 ± 19 kcal vs. 426 ± 19 kcal; *p* = 0.2), but there was a significant decrease in the overall energy density when fruit and vegetable portion was doubled (from 0.95 ± 0.02 to 0.89 ± 0.02 kcal/g; *p* = 0.005). Furthermore, girls had significantly less total energy intake than boys when served large fruit and vegetable portion (*p* = 0.01). Food preferences may moderate the effect of portion size and fruit & vegetable intake.	**Strengths:** (1) Adjusting fruit and vegetable intake analysis with preference and liking of foods; (2) Familiarisation of setting for children before testing; (3) Children conditioning before intervention: abstain from food and drinks for 2 h. **Limitations:** (1) Using puree fruit (apple sauce) instead of solid fruit which may influence preference and satiety; (2) Narrow age range of children and exclusion of children who disliked most foods which limits generalisability of findings; (3) Homogenous study sample (majority were Black or African American).
⑥ DiSantis et al., 2013 [24] **Title:** “Plate size and children’s appetite: Effects of larger dishware on self-served portions and intake” **Location:** Philadelphia, Pennsylvania **Design:** Within-subjects experimental design. Randomisation of feeding condition order.	**Objective:** (1) Determine dishware size effects on self-served portion sizes and energy intake of young children; (2) Identify children’s characteristics who responded to more food with larger dishware. **Sample size:** 42 first-grade elementary students (exact age was not given). **Intervention method:** Children served themselves using either child or adult-size dishware (double in size and volume) in a buffet type line. Entrée of amorphous (pasta with meat sauce) and unit form (chicken nuggets) were served on separate days, whereas fruit and vegetable were self-served at all meals. Bread and milk were served in fixed portions. **Eating setting:** Lunch time in school classroom.	**Intervention period:** Once a week for 8 weeks **Outcome evaluation:** Energy intake (kcal) of each meal, food-liking assessment. **Follow-up:** No follow up (intake was estimated immediately after each meal)	There was a difference in total energy intake between the dishware sizes (*p* = 0.002). On average, children consumed 90.1 kcal (SE = 29.4 kcal) more when using adult-size dishware, and this effect was seen in 80% of children. Type of entrée also influenced the total energy intake, where children had higher energy intake when unit entrée was offered compared to amorphous entrée (*p* = 0.001). Based on each type of food, children served themselves more entrée (+57.6 kcal, SE = 19.7 kcal) and more fruit (+15.7 kcal, SE = 6.3 kcal) when using adult-size than child-size dishware (*p* < 0.05), but there was no effect on vegetable intake. Furthermore, food insecurity was a significant predictor of large portion size (*p* = 0.04), whereas sex and body mass index (BMI) showed no significant association.	**Strengths:** (1) Natural eating setting (school lunch); (2) Familiarisation of procedure for children before intervention; (3) Analysis models were adjusted for entrée type, child’s sex and weight status, food insecurity, and child’s food-liking. **Limitations:** (1) Randomisation of condition order only in the classroom level, not for individual children; (2) Children’s behaviour of self-serving food may vary; (3) Homogenous study sample (majority were African American).
⑦ Kral et al., 2014 [25] **Title:** “Role of child weight status and the relative reinforcing value of food in children’s response to portion size increases” **Location:** Philadelphia, Pennsylvania **Design:** Randomised crossover design	**Objective:** To compare energy intake at a meal between children with normal-weight and children with obesity when the portion size of energy-dense foods was increased. **Sample size:** 50 children (25 normal-weight, 25 obese) aged 8–10 years old. **Intervention method:** Children were served dinner meal (chicken nuggets, hash browns, ketchup, green beans, brownies) which varied across 3 portion sizes (100%, 150%, 200%). **Eating setting:** Dinner in “Centre for Weight and Eating Disorders at University of Pennsylvania”.	**Intervention period:** Once a week for 3 weeks. **Outcome evaluation:** Energy intake (kcal) at each meal, taste preference of all foods given. **Follow-up:** No follow-up (intake was estimated immediately after each meal).	Total energy intake (mean ± SD) across the 100, 150, and 200% portion sizes, with children’s weight groups combined, were 921 ± 40, 1046 ± 41, and 1041 ± 40 kcal, respectively. While there was a trend suggesting that children with obesity were more responsive to portion size changes, this difference was not statistically significant (*p* = 0.1), meaning that children in all weight status consumed more calories in response to larger portion sizes (>50–60% of estimated energy requirements).	**Strengths:** (1) Assessing multiple components of children’s response to portion size; (2) Familiarisation and children’s conditioning: abstain from food and drinks for 2 h before testing. **Limitations:** (1) Relatively small sample size; (2) Conducted in laboratory setting; (3) No follow up or longitudinal data collected; (4) Children’s variation in hunger level which may impact intake at meals.
⑧ Mooreville et al., 2015 [26] **Title:** “Individual differences in susceptibility to large portion sizes among obese and normal-weight children” **Location:** Philadelphia, Pennsylvania **Design:** Within-subjects design, with randomisation of feeding condition sequences.	**Objective:** To investigate the association of young children’s vulnerability to large portion sizes of foods with weight status and appetite regulation traits. **Sample size:** 100 children (66 with normal weight, 34 with obesity) aged 5–6 years. **Intervention method:** Children were served a dinner meal containing pasta, corn, applesauce, cookies, and 2% milk. All foods (except milk) varied across 4 portion size conditions: 100% (677 kcal), 150% (1015 kcal), 200% (1353 kcal), or 250% (1691 kcal). **Eating setting:** Dinner time in laboratory.	**Intervention period:** Once a week for 4 weeks. **Outcome evaluation:** Energy intake (kcal) of each meal, food preference, food responsiveness, satiety responsiveness, and enjoyment of food. **Follow-up:** No follow-up (intake was estimated immediately after each meal).	Children consumed higher total energy intake with increasing portion sizes across all conditions (*p* < 0.001). On average, children consumed 479.9 ± 167.8 kcal in 100% portion size condition, 531.2 ± 204.9 kcal in 150% condition, 540.9 ± 216.5 kcal in 200% condition, and 585.1 ± 247.8 kcal in 250% condition. The effect of portion size condition on total energy intake did not vary with children’s weight status (*p* = 0.6) but varied with satiety responsiveness (*p* < 0.05) and food responsiveness (*p* = 0.05).	**Strengths:** (1) Assessing individual variation in children’s susceptibility of large portion size; (2) Relatively high sample size; (3) Children’s conditioning: fasting for 2 h before testing. **Limitations:** (1) Homogenous study sample (all children were non-Hispanic black children, majority from low-income families); (2) No follow up or longitudinal data collected; (3) Laboratory setting; (4) Imbalanced proportion of children with normal weight and obesity.
⑨ Kerr et al., 2019 [27] **Title:** “Child and adult snack food intake in response to manipulated pre-packaged snack item quantity/variety and snack box size: a population-based randomised trial” **Location:** Australia (multi-cities) **Design:** Population-based randomised trial	**Objective:** To investigate how intake is affected by modification of pre-packaged snack food in terms of item quantity and variety dishware (boxed container) size. **Sample size:** 1299 children (11–12 years old) and 1274 parents from the Longitudinal Study of Australian Children (LSAC). **Intervention method:** Snack food items (savoury crackers, sweet biscuits, milk chocolate, cheese, muesli/granola bars, wheat fruit bites, and peaches in juice) were presented in one of four conditions: (1) small box, fewer items, (2) larger box, fewer items, (3) small box, more items, and (4) larger box, more items. **Eating setting:** “Food Stop” (snacking area) in the assessment centre.	**Intervention period:** One session between February 2015 and March 2016 (14 months). **Outcome evaluation:** Food intake (g) and energy intake (kJ) of snacks, hunger scale before eating. **Follow-up:** No follow-up (intake was estimated immediately after each meal).	Children’s intake increased by 10 g (95% CI 3–17 g) and 349 kJ (95% CI 282–416 kJ) when offered more variety/quantity of snacks (*p* < 0.01). Box size did not affect intake (*p* = 0.7, for grams consumed; *p* = 0.5, for kilojoules consumed). Based on children’s characteristics, boys consumed significantly more energy (kJ), but not amount in grams, of snacks compared to girls (*p* < 0.001). However, neither energy (kJ) nor consumption in grams differed by children’s weight and social-economic status. In adults, neither box size nor variety or quantity of snack items had significant effect on consumption.	**Strengths:** (1) Large sample size (population-based); (2) Although tightly controlled, the break time setting was more realistic than laboratory setting; (3) Adults and children were exposed to identical but separate manipulations. **Limitations:** (1) Distraction of other activities (e.g., reading) while the study participants having snacks; (2) Adults exposed to the information booklet prior testing which mention that food intake would be evaluated; (3) Children had fasted longer and were hungrier than adults at the Food Stop.
⑩ McGale et al., 2020 [28] **Title:** “The influence of front-of-pack portion size images on children’s serving and intake of cereal” **Location:** UK **Design:** Between-subjects design, with randomisation of participants to intervention condition.	**Objective:** To investigate the effect of front-of-package portion size image on children self-served portions and consumption. **Sample size:** 41 children aged 7–11 years old. **Intervention method:** Children were exposed to cereal box condition depicting either a small visual cue of recommended cereal serving size (30 g), or a larger and more typical front-of-pack portion size (90 g). **Eating setting:** Breakfast meal in laboratory.	**Intervention period:** One session between February and November 2015 (10 months). **Outcome evaluation:** Cereal serving and intake (gram), children’s perception of the portion size, hunger scale. **Follow-up:** No follow-up (intake was estimated immediately after each meal).	Children served themselves (+7 g, 37%) and consumed (+6 g, 63%) more cereal when exposed to larger-portion-size visual cue compared to small-portion-size visual cue (*p* = 0.015 for cereal served, *p* = 0.002 for cereal consumed). The total meal consumed (cereal and milk) was higher in large compared to small portion size visual cue (mean ± SE: 73.4 ± 7.4 g vs. 47.1 ± 7.8 g). Furthermore, 63% of children accepted the image of portion as appropriate, regardless of condition, meaning that children might be susceptible to manipulations of external cues.	**Strengths:** (1) Using natural control of large portion size package which represent typical cereal package on the market; (2) The analysis accounted covariates (BMI, age, sex, hunger rating). **Limitations:** (1) Relatively small sample size; (2) Did not assess the effect on energy intake; (3) Fixed amount of milk provided (100 g) which create ceiling effect.

^1^. Statistical significance, mean difference, and variance were given when available.

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
