# Peer review of "The Effect of Portion Size Interventions on Energy Intake and Risk of Obesity in School-Aged Children: A Systematic Review and Meta-Analysis"

_nutrients, 2025, doi:10.3390/nu17182911_

Round 1
Reviewer 1 Report
Comments and Suggestions for Authors
This was a very well written and presented review of the impact of portion size variation on children's energy consumption. The findings are very clearly presented, and the findings are clearly situated in previous research. My comments mainly reflect the need for greater acknowledgement of the results relating to single meals only, which is clear in some places but I felt was needed elsewhere to ensure there was not too much significance claimed for the type of studies analysed.
Specific comments:
Abstract:
- This was very clear, but for transparency it should reflect that the studies were looking at experimental manipulations for a single meal only.
Introduction:
- Line 62 is confusing – the link between the two clauses is unclear. Having looked at the systematic review referred to (no 14), there is very little overlap because of the different aims (observed vs manipulated portion sizes). Adding more specific detail on this would support your case for why this study is needed given this other recent publication.
- Line 71 - It would be helpful to be specific in this paragraph about the age of your focus, rather than “this age”. At one point you argue (or appear to) that your study is different from those that have looked at adolescents, yet your age range includes adolescents. Clarifying this would be useful.
Discussion:
- Line 272-275 – I did not follow the logic of this argument; typically highly palatable fast food is presented as something that we commonly eat more of than we need, and hence would expect to be more responsive to portion size. The alternative therefore needs to be spelt out more.
- Section 4.2 – The discussion here risks overclaiming to me - as it assumes that children do not compensate for over-eating in the target meal, by eating less in subsequent meals. There is some discussion of this in a later section - but this caveat is needed here, as to this point in the Discussion there is no acknowledgement of this limitation of what we can conclude from the findings. Variations in self-regulation following overeating have been reported previously, so needs to be part of this discussion to avoid appearing to either over-simplify or overclaim in these interpretations.
- The potential for variation in compensatory response according to different characteristics may also be relevant to section 4.3 when speculating on the responses of children with different body weights etc.
Author Response
- Comments 1: This was very clear, but for transparency it should reflect that the studies were looking at experimental manipulations for a single meal only.
- Response 1: The reviewer’s suggestion has been added to the conclusion of the abstract: ‘However, this finding was limited by being based mainly on studies which manipulated portion size at a single meal, in a laboratory setting, and with only short-term measures of energy intake’
- Comments 2: Line 62 is confusing – the link between the two clauses is unclear. Having looked at the systematic review referred to (no 14), there is very little overlap because of the different aims (observed vs manipulated portion sizes). Adding more specific detail on this would support your case for why this study is needed given this other recent publication.
- Response 2:
Thank you for this helpful comment. The section has been rewritten and simplified as suggested:
‘When offered a larger portion size, children and adults increase their energy intake, a phenomenon known as the portion size effect [10]. For example, systematic reviews in adults [11] and young children [12] found that larger portion sizes were associated with greater energy intake,. However, a similar review has not been conducted in school-aged children who may be particularly vulnerable to the portion size effect [13]. Whether larger portions affect the risk of obesity is uncertain. However, a recent systematic review showed associations between portion size and childhood adiposity [14], although in contrast to the current study, this previous review was a narrative synthesis and based mainly on observational evidence[14]. Therefore, interventions that reduce portion size have been recommended to recalibrate consumption norms and to help prevent obesity [15].
- Response 2:
- Comments 3: Line 71 - It would be helpful to be specific in this paragraph about the age of your focus, rather than “this age”. At one point you argue (or appear to) that your study is different from those that have looked at adolescents, yet your age range includes adolescents. Clarifying this would be useful.
- Response 3:
‘This age’ has been changed to children 5-17 years.
We agree that inclusion of the word ‘adolescents’ in this section is confusing and so the section has been rewritten with the precise age given as below:
‘School age is an important period in shaping health behavior in children [18], and portion sizes of energy-dense foods and soft drinks have been associated with higher BMI in school-aged children [13]. However, the effect of portion size interventions on food and energy intake has not been systematically investigated in school-aged children (5-17 years).
- Response 3:
- Comments 4: Line 272-275 – I did not follow the logic of this argument; typically highly palatable fast food is presented as something that we commonly eat more of than we need, and hence would expect to be more responsive to portion size. The alternative therefore needs to be spelt out more.
- Response 4:
Yes - we agree. We made the statement that the lack of an association between portion size and energy intake in this particular study was due to the palatability of the foods based on the conclusions of the study authors. However, the reason for the lack of association between portion size and energy intake in this study is not really known and hence we have deleted the statement ‘because it was based on highly palatable foods.’ [line 278-280]
- Response 4:
- Comments 5: Section 4.2 – The discussion here risks overclaiming to me - as it assumes that children do not compensate for over-eating in the target meal, by eating less in subsequent meals. There is some discussion of this in a later section - but this caveat is needed here, as to this point in the Discussion there is no acknowledgement of this limitation of what we can conclude from the findings. Variations in self-regulation following overeating have been reported previously, so needs to be part of this discussion to avoid appearing to either over-simplify or overclaim in these interpretations.
- Response 5:
We have discussed the possibility of compensation later in the discussion. However, as suggested by the reviewer, we have also included this caveat in the first sentence of this paragraph.
‘Although further research is required to investigate if children compensate for over-eating at one meal by eating less in subsequent meals, the 86.0 kcal increase in energy intake per meal, when children are offered a larger portion compared to the reference, if sustained, may contribute to excess weight gain.’
We have also toned down the conclusion section and use the word may rather than likely to. [line 475]
- Response 5:
- Comments 6: The potential for variation in compensatory response according to different characteristics may also be relevant to section 4.3 when speculating on the responses of children with different body weights etc.
- Response 6:
We completely agree and hence have added another sentence to section 4.3
‘Thus, while the portion size effect was generally consistent across weight status in children, there might be differences in individual appetitive traits that influence the response to different portion sizes, a hypothesis requiring further investigation. Furthermore, whether weight status affects the variability in any compensatory response to larger portions also requires further research.’ [line 374-379]
- Response 6:
Reviewer 2 Report
Comments and Suggestions for Authors
The manuscript addresses a relevant and timely public health question—the effect of portion size interventions on energy intake in school-aged children. The systematic review and meta-analysis are well structured, follow PRISMA guidelines, and provide a clear synthesis of experimental studies. The paper is generally well written, with transparent methodology, a comprehensive search strategy, and a clear presentation of results. The findings have practical implications for obesity prevention strategies. However, several points raised my attention:
- The I² value of 91% is substantial. While the authors acknowledge heterogeneity, further exploration through meta-regression (e.g., age range, intervention type, setting, type of food, sample size) would be valuable to better understand sources of variation. Alternatively, more granular subgroup analyses beyond “laboratory vs. natural setting” could be added.
- The paper emphasizes that none of the included studies assessed long-term changes in BMI or habitual energy intake. This is an important limitation—consider recommending that future research incorporate repeated measures and follow-up periods to assess sustained effects and weight outcomes.
- A short section on whether repeated exposure to smaller portions can recalibrate norms over time would enrich the implications.
- Although grey literature was excluded, the potential for publication bias is not formally evaluated. A funnel plot or Egger’s test could be conducted if data points are sufficient. If not possible, this should be stated explicitly.
- Most studies were from high-income, Western countries (primarily the USA). The authors should discuss more explicitly how cultural and food environment differences could limit extrapolation to other contexts.
- It would be helpful to clearly state whether effect sizes were weighted by study variance in the meta-analysis and whether random-effects models accounted for within-study correlation in crossover designs.
Minor points
- Check for consistency in terminology (sometimes “larger portion size” vs. “portion size intervention”).
- In Table 2, some entries do not specify exact ages or intervention periods—adding these would improve completeness.
- Consider tightening some sections of the discussion that repeat numerical results already presented in the results section.
- Ensure that all abbreviations (e.g., BMI, QCC) are defined at first mention in the main text and tables.
Author Response
- Comments 1: The I² value of 91% is substantial. While the authors acknowledge heterogeneity, further exploration through meta-regression (e.g., age range, intervention type, setting, type of food, sample size) would be valuable to better understand sources of variation. Alternatively, more granular subgroup analyses beyond “laboratory vs. natural setting” could be added.
- Response 1:
We completely agree that there is high heterogeneity and have clarified this further under the study limitations on page 17.
‘Second, there was high heterogeneity between studies because of the many different portion-size interventions used, the different foods manipulated and different participant populations.’ [line 448-450]
However, with only 10 studies (and 8 showing a portion size effect) we feel that the sample is not large enough to investigate factors that could account for this heterogeneity using meta-regression. Similarly, we feel that there is too much variation in age, intervention type, or food group manipulated to allow sufficiently large and homogenous groups to allow further meaningful subgroup analyses. Furthermore, according to the Cochrane Handbooks, meta-regression should generally not be considered when there are fewer than ten studies in a meta-analysis. (reference: https://www.cochrane.org/authors/handbooks-and-manuals/handbook/current/chapter-10)
- Response 1:
- Comments 2: The paper emphasizes that none of the included studies assessed long-term changes in BMI or habitual energy intake. This is an important limitation—consider recommending that future research incorporate repeated measures and follow-up periods to assess sustained effects and weight outcomes.
- Response 2:
The conclusions have been revised as suggested by the reviewer:
‘Our systematic review suggests that larger portion sizes led to higher energy intake in school-aged children with an effect size (86 kcal/meal), that may contribute to increase in energy intake and hence risk of obesity. Children appear susceptible to the portion size effect regardless of weight status, although further investigation is required to understand the influence of sex, age, and socioeconomic status. Furthermore, future research needs to incorporate repeated measures and longer follow-up periods to assess sustained effects on habitual energy intake and long-term changes in BMI. While the generalisability of the evidence should be interpreted cautiously, given the limitations of previous research in this area, this review suggests that portion-size interventions could help reduce the risk of obesity in school-aged children’
Furthermore, the following is included at the end of the first paragraph in the discussion on page 14.
‘Due to the lack of longer-term follow-up, insufficient evidence was available to determine if portion size interventions could affect children's overall or habitual energy intake, or lead to a change in body weight and risk of obesity’
- Response 2:
- Comments 3: A short section on whether repeated exposure to smaller portions can recalibrate norms over time would enrich the implications.
- Response 3: As this review focused on the effect of intervention between larger portion compared to normal (reference) portion size, we think that results of repeated exposure to smaller portions in recalibrating consumption norms cannot be investigated by the current review
- Comments 4: Although grey literature was excluded, the potential for publication bias is not formally evaluated. A funnel plot or Egger’s test could be conducted if data points are sufficient. If not possible, this should be stated explicitly.
- Response 4:
A section has been added to assess publication bias (section 3.6, page 13).
‘The potential for publication bias was assessed by generating funnel plots of nine studies included in the meta-analysis, where symmetry can be observed (Figure 4). The test for asymmetry from Egger’s test revealed non-significant result (t = -0.9, p = 0.3), which further showed that potential bias was not from publications reported’
- Response 4:
- Comments 5: Most studies were from high-income, Western countries (primarily the USA). The authors should discuss more explicitly how cultural and food environment differences could limit extrapolation to other contexts.
- Response 5:
The above suggestion has been incorporated in to the text in the section on limitations as below (page 18):
‘Fourth, most studies were conducted in the United States, with only one in the United Kingdom and one in Australia. Thus, differences in culture and food environment (e.g. frequency and timing of eating, and availability and consumption of highly palatable snack foods) could limit the generalisability of findings to other regions. This is especially important because few portion size interventions have been conducted in low- or middle-income countries which face a rapid increase in obesity rates’
- Response 5:
- Comments 6: It would be helpful to clearly state whether effect sizes were weighted by study variance in the meta-analysis and whether random-effects models accounted for within-study correlation in crossover designs.
- Response 6:
The above suggestion has been incorporated into section 2.5 (page 4):
‘Random-effects meta-analysis with inverse-variance weighting was performed with Review Manager (RevMan) 5.4 and R with “meta” package. Results were presented as the mean difference with 95% confidence intervals (CI) in energy intake (kcal) between portion size conditions (reference vs. larger portion offered).’
- Response 6:
- Comments 7: Check for consistency in terminology (sometimes “larger portion size” vs. “portion size intervention”).
- Response 7:
Thank you for this very helpful comment. We have revised the manuscript to make clear that portion size intervention is only used as a generic term and where we specifically mean ‘larger portions’ the manuscript has been changed to say this. For example in the following sections:.
Abstract ‘Only one study found that the larger portion sizes did not affect energy intake, ….’
Section 3.4: ‘Only one study, based on altering portions and eating rate of a fast food meal, found that larger portion sizes did not lead to a statistically significant effect on energy intake [20].,….’
Section 4.2: Importantly, the effect of larger portion sizes was similar in natural and laboratory settings (which minimizes confounding factors that affect eating behavior).
- Response 7:
- Comments 8: In Table 2, some entries do not specify exact ages or intervention periods—adding these would improve completeness.
- Response 8:
Table 2 has been checked and where available the missing data has been added
- Response 8:
- Comments 9: Consider tightening some sections of the discussion that repeat numerical results already presented in the results section.
- Response 9:
The following sentences have been removed from the discussion
Page 13, 'which included those with a randomised crossover design (n=3), within-subject design (n=3), between-subject design (n=3), and a population-based randomised controlled trial (n=1)'.
Page 14, '63% of children perceived the portion image as the right amount to eat regardless of whether the smaller or larger portion size image was on the box'
- Response 9:
- Comments 10: Ensure that all abbreviations (e.g., BMI, QCC) are defined at first mention in the main text and tables.
- Response 10: Both BMI and QCC are defined in the first mention in the main text and abstract, and also now defined in Tables 1 and 2.
Round 2
Reviewer 2 Report
Comments and Suggestions for Authors
I have no further comments for the authors